# Adverse Cardiovascular Events in Non-Traumatic Intracranial Hemorrhage and Ischemic Stroke Survivors

**DOI:** 10.3390/jcm11236885

**Published:** 2022-11-22

**Authors:** Marco Pasi, Grégoire Boulouis, Arnaud Bisson, Julien Herbert, Alexandre Bodin, Charlotte Cordonnier, Gregory Y. H. Lip, Laurent Fauchier

**Affiliations:** 1Neurology Department, CHU de Tours et Université François Rabelais, 37000 Tours, France; 2Neuroradiology Department, CHU de Tours et Université François Rabelais, 37000 Tours, France; 3Service de Cardiologie, CHU Trousseau et Université François Rabelais, 37000 Tours, France; 4Inserm U1172—LilNCog—Lille Neuroscience & Cognition, CHU Lille, Université de Lille, 59000 Lille, France; 5Liverpool Centre for Cardiovascular Science at University of Liverpool, Liverpool John Moores University and Liverpool Heart & Chest Hospital, Liverpool L14 3PE, UK; 6Department of Clinical Medicine, Aalborg University, 9100 Aalborg, Denmark

**Keywords:** ischemic stroke, intracranial hemorrhage, atrial fibrillation, cardiovascular death, heart failure, myocardial infarction, major cardiovascular event

## Abstract

Background: We aimed to evaluate different measures of adverse cardiovascular events between non-traumatic intracranial hemorrhage (ICrH) and ischemic stroke (IS) survivors with and without atrial fibrillation (AF). Methods: Using a national hospitalization database we compared incidences and risks of all-cause in-hospital death, cardiovascular death, non-cardiovascular death, MACE-HF (in-hospital cardiovascular death, myocardial infarction, IS or new-onset heart failure [HF]) between ICrH and IS survivors with and without AF. Propensity-score matching was also performed. Results: We identified 40,523 survivors of IS and 12,028 survivors of an ICrH without AF, and 20,449 IS survivors and 3574 ICrH survivors with AF. In unadjusted, adjusted and matched analyses, ICrH patients without AF had a higher risk of all-cause in-hospital death (Hazard Ratio (HR; for matched analyses) 1.80; 95% confidence interval (CI) 1.74–1.86), cardiovascular death (HR; 2.79; CI 2.64–2.94), MACE-HF (HR 1.97; CI 1.89–2.06) and new cerebrovascular events (HR 1.75; CI 1.57–1.96) but with lower risk of myocardial infarction (HR 0.81; CI 0.70–0.94), major bleeding (HR 0.92; CI 0.87–0.98) and new onset HF (HR 0.85; CI 0.79–0.91) compared to IS survivors without AF. Similar results were found for ICrH and IS survivors with AF except for myocardial infarction (HR 1.05; CI 0.79–1.34) and new onset HF (HR 0.94; CI 0.84–1.06) that were similar between the two groups. Conclusions: Adverse cardiovascular events are more frequent in ICrH survivors compared to IS survivors. New onset HF is a relatively frequent event after ICrH, especially in those patients with comorbid atrial fibrillation.

## 1. Introduction

Stroke is a highly incident and prevalent disease associated with a relevant burden in terms of mortality and disability [1]. According to a recent report of the global burden of diseases around 60% of all new strokes worldwide are ischemic, almost 30% hemorrhagic and the remaining almost 10% is constituted by subarachnoid hemorrhage [1].

Ischemic stroke (IS) and non-traumatic intracranial hemorrhage (ICrH) share common risk factors that influence the risk of subsequent vascular events, both ischemic and hemorrhagic [2]. Several reports indicate that IS survivors, irrespective of the underlying etiology, are associated with an elevated risk of recurrence and other arterial ischemic events [3,4]. Population based and hospital-based cohorts suggest that also ICrH survivors are at greater risk of thromboembolic events in the long-term [5,6,7]. Furthermore, patients with ICrH share an elevated risk of recurrence that can reach almost 10% per year in lobar ICrH patients with cerebral amyloid angiopathy [8,9].

Atrial fibrillation (AF) is frequent (up to 24%) and it is likely a risk factor for IS in ICrH survivors [6]. In this group of patients, oral anticoagulation might be warranted and the optimal preventive strategy in ICrH survivors with comorbid AF requires further evaluation. In a recent a pooled analysis of two population-based studies including 674 first-ever intracerebral hemorrhage patients, comorbid AF was associated with a more than eight-fold increased risk of IS and a doubling of the risk of any serious vascular event [6]. Furthermore, the subgroup of patients with a history of occlusive vascular disease had even higher rates of adverse cardiovascular event [6].

The occurrence of new vascular events and the influence of AF on adverse cardiovascular outcome have never been directly compared between IS and ICrH survivors. In this study, we aimed to compare the incidence and the risk of different measures of adverse cardiovascular events between ICrH and IS with or without comorbid AF.

## 2. Methods

Using the national hospitalization database covering hospital care from the entire French population, we performed a retrospective longitudinal cohort study. Data for all patients admitted in French hospitals from January to December 2013 with at least 5 years of complete follow-up (or earlier if death) were collected from the national administrative PMSI (Programme de Médicalisation des Systèmes d’Information) database, as previously described [10,11]. As patients were not involved in its conduct, there was no impact on their care. For this specific study ethical approval was not required [10]. The French Data Protection Authority granted access to the PMSI data.

### 2.1. Study Population

From 1 January 2013 to 31 December 2013, 3,381,472 patients (age ≥ 18 years) were hospitalized and had at least 5 years of complete follow-up (or earlier if suffered death). Patient characteristics such as demographics, past medical history and events during hospitalization or follow-up was described using data collected in the hospital records. For each hospital stay, combined diagnoses at discharge were obtained. Each variable was identified using International Classification of Diseases, Tenth Revision (ICD-10) codes. We identified all patients with a history of IS (ICD-10 code I63) and those with ICrH (I61, I62 and their subsections). We also identified history of atrial fibrillation (AF) (I48).

### 2.2. Outcomes

We evaluated the incidence of all-cause in-hospital death, cardiovascular death and non-cardiovascular death, major cardiovascular events (MACE-HF, i.e., in-hospital cardiovascular death, MI, IS or new onset heart failure (HF)), major bleeding and new type of cerebrovascular event (i.e., IS for ICrH survivors and ICrH for IS survivors).

The endpoints were evaluated using follow-up data starting from the date of first hospitalization until the date of each specified outcome or date of last news in the absence of the outcome. Outcomes were identified using their respective ICD-10 and the mode of death (cardiovascular or non-cardiovascular) was identified based on the main diagnosis during hospitalization resulting in death. The list of ICD-10 codes used for outcome is reported in the Appendix A.

### 2.3. Statistical Analysis

Binary baseline characteristics were described as frequency and percentages and continuous variable as means (standard deviations (SD)). Multivariable analyses were performed using a Cox model with all baseline characteristics and hazard ratio (HR) was reported. The model by Fine and Gray was also used for competing risks for (1) cardiovascular and non-cardiovascular death and (2) other clinical events and all-cause death, reporting sub-distribution hazard ratios (sHR).

In order to account for the presence of significant differences in baseline characteristics, propensity-score matching was also performed. Statistical methods used for propensity-score matching are detailed in Appendix A and Appendix A.

Statistical significance was taken at *p* < 0.05. All analyses were performed using Enterprise Guide 7.1, (SAS Institute Inc., SAS Campus Drive, Cary, NC, USA) and STATA version 16.0 (Stata Corp, College Station, TX, USA).

## 3. Results

In 2013, we identified 40,523 survivors of IS and 12,028 survivors of an ICrH without history of AF, and 20,449 IS and 3574 ICrH survivors with comorbid AF (flow chart in Appendix A). The mean time between IS and the date of inclusion was 358 ± 410 days. The mean time between ICrH and date of inclusion was 256 ± 380 days. The mean time between diagnosis of AF and date of inclusion was 469 ± 430 days.

### 3.1. Population without AF

Table 1 shows univariate comparisons between IS and ICrH survivors without AF. Differences between these two groups after propensity matching are reported in Appendix A. Incident outcomes in the matched population with no AF according to ICrH or IS are reported in Table 2. In the matched population, ICrH survivors had higher incidences of all-cause in-hospital death, cardiovascular death and non-cardiovascular death, MACE-HF and new type of cerebrovascular event compared to IS survivors. Conversely, IS patients had higher annual incidences of myocardial infarction, major bleeding and new onset HF. In adjusted, unadjusted and matched analyses (Table 3), ICrH survivors without AF had higher risk of all-cause in-hospital death (all reported results refer to matched analyses; HR 1.80, CI 1.74–1.86), cardiovascular death (HR 2.79; CI 2.64–2.94), non-cardiovascular death (HR 1.28; CI 1.22–1.34), MACE-HF (HR 1.97; CI 1.89–2.05), and new type cerebrovascular events (HR 1.75; CI 1.57–1.96) but lower risk of myocardial infarction (HR 0.81; CI 0.70–0.94), major bleeding (HR 0.92; CI 0.87–0.98) and new onset HF (HR 0.85; CI 0.79–0.91) compared to IS survivors. Results of competing risk analyses and sHR are shown in Appendix A.

In Figure 1 are shown the cumulative incidences for all-cause in-hospital death, cardiovascular death, myocardial infarction, new onset HF, new type of cerebrovascular event and MACE-HF during follow-up in the matched populations with no AF.

### 3.2. Population with AF

Table 1 shows univariate comparisons between IS survivors and ICrH survivors with AF. Differences between these two groups after propensity matching are reported in Appendix A. In the matched population, ICrH survivors with AF had higher rates of all-cause in-hospital death, cardiovascular death, non-cardiovascular death, MACE-HF and new type of cerebrovascular event but lower incidence of major bleeding compared to IS survivors (Table 2). Ischemic stroke and ICrH survivors with AF shared similar incidences of myocardial infarction and new onset HF (Table 2). In unadjusted, adjusted and propensity matched analyses (Table 4), ICrH survivors with comorbid AF had a higher risk of all-cause in-hospital death (all reported results refer to matched analyses; HR 1.56; CI 1.47–1.64), cardiovascular death (HR 2.08; CI 1.91–2.25), non-cardiovascular death (HR 1.21; CI 1.13–1.31), MACE-HF (HR 1.71; CI 1.60–1.83) and new type of cerebrovascular event (HR 2.29; CI 1.93–2.73) but with similar risk of myocardial infarction (HR 1.05; CI 0.79–1.39), and new onset HF (HR 0.94; CI 0.84–1.06) compared to IS survivors. Results of competing risk analyses and sHR are shown in Appendix A.

Figure 2 shows the cumulative incidences for all-cause in-hospital death, cardiovascular death and myocardial infarction, new onset HF, new type of cerebrovascular event and MACE-HF during follow-up in the matched populations with AF.

## 4. Discussion

The overall burden associated with cardiovascular events after ICrH is higher compared to IS survivors. This holds true for patients with and without comorbid AF. We found that new onset HF is a not negligible event after ICrH, especially in those patients with comorbid AF. Our results highlight an urgent need for optimization of cardiovascular preventive measures in ICrH survivors.

In our large dataset, we found that ICrH survivors had almost a 2-fold increase in all-cause in-hospital death and close to a 3-fold increase in in-hospital cardiovascular death compared to IS stroke survivors. Previous studies reported high rates of long-term mortality, with a survival rate at 10 years between 24 and 38% [12]. In our study the annual incidence of in-hospital death was 25.2%/year (CI 24.7–25.8) in ICrH survivors without AF and the incidence almost doubled (40.5%/year; CI 39.1–42.0) in those with AF. This was mostly due by an increase in cardiovascular death in ICrH survivors with comorbid AF. Our findings confirm, specifically in ICrH survivors, that AF is a strong risk factor for mortality [13].

In our dataset, we found that ICrH survivors had almost a two-time increased risk of MACE-HF compared to IS stroke survivors in both group with and without AF. This finding might be in part related to the high in-hospital mortality associated with new cardiovascular event in ICrH survivors. A relatively high percentage of ICrH survivors are whether functional impaired just after the event or they develop a functional decline during the follow-up [14]. In this already frail population, one additional cardiovascular event might likely be fatal.

Due to the concomitant high bleeding and thromboembolic risk, the optimal preventive strategy in ICH survivors at high risk of thromboembolic events is still a clinical dilemma [15,16]. Therefore, our objective was to specifically differentiate patients according to the presence of AF, a major risk factor for arterial occlusive events. In our analyses, we found that ICrH survivors in both groups (AF and no AF) share an overall worst cardiovascular outcome profile than IS survivors according to different adverse outcome measures. Furthermore, in ICrH survivors with AF incidences of myocardial infarction and new onset heart failure were similar to those of IS survivors.

New onset HF has been poorly evaluated in ICrH patients so far. Our results suggest that new onset HF is a non-negligible event after ICrH with an incidence of approximately 4%/year in patients without AF patients that almost double in AF patients. Heart failure occuring after ICrH might in part contribute to the significant risk of IS in ICrH survivors.

Finally, we found that major bleeding, irrespective of the presence of AF, were higher in IS survivors. Our finding might be in part related to a more widespread use of antithrombotic drugs in this population. Nevertheless, this higher risk of major bleeding in IS survivors was not associated with a higher mortality compared to ICrH survivors. Conversely, ICrH survivors are less likely to receive antithrombotic therapy as clinicians might be afraid of bleeding. In line with this hypothesis, we also found that the risk of IS in ICrH survivors is almost doubled compared to the risk of subsequent ICrH in IS survivors.

### Limitations

We acknowledge that in our strategy of search of ICrH we used the codes I61, I62 and their subsections. This means that we have potentially included also other type of ICrH such as non-traumatic subdural hematoma.

One of the major limitations of our study is that it was based on administrative data. The PMSI database contains diagnoses using ICD-10 codes, which are obtained at hospital discharge and are the physician’s responsibility; the data were not systematically checked externally, and this could have created an information bias. However, the large scale of the database is likely to compensate for some of the possible biases, and, as the coding of complications is linked to reimbursement and is regularly controlled, it is also likely to be of high quality. Our analysis on mortality is limited to deaths that occurred during hospitalization, and the results cannot be generalized to patients who died before hospitalization or after discharge. However, French data show that hospital deaths account for 66% of all deaths in patients aged 70–80 years, similar to the mean age of our population [17]. Our study, therefore, provides important insights into the reasons why patients with IS and ICrH die. Furthermore, we cannot exclude that some patients had a previous stroke in a period not covered by the database, but we do not think that markedly influenced our results. Finally, we are not reporting the risk of recurrent ICrH. This clinically relevant outcome of interest has already been largely evaluated in previous reports [8,9].

In conclusion, our paper found that ICrH survivors are at high risk of major adverse cardiovascular outcomes. This risk was higher compared to IS survivors. ICrH survivors, especially those with AF, shared a not negligible risk of new onset heart failure. Our results highlight an urgent need for optimization of treatment strategies for the prevention of major adverse cardiovascular event in ICrH survivors.

## Figures and Tables

**Figure 1 jcm-11-06885-f001:**
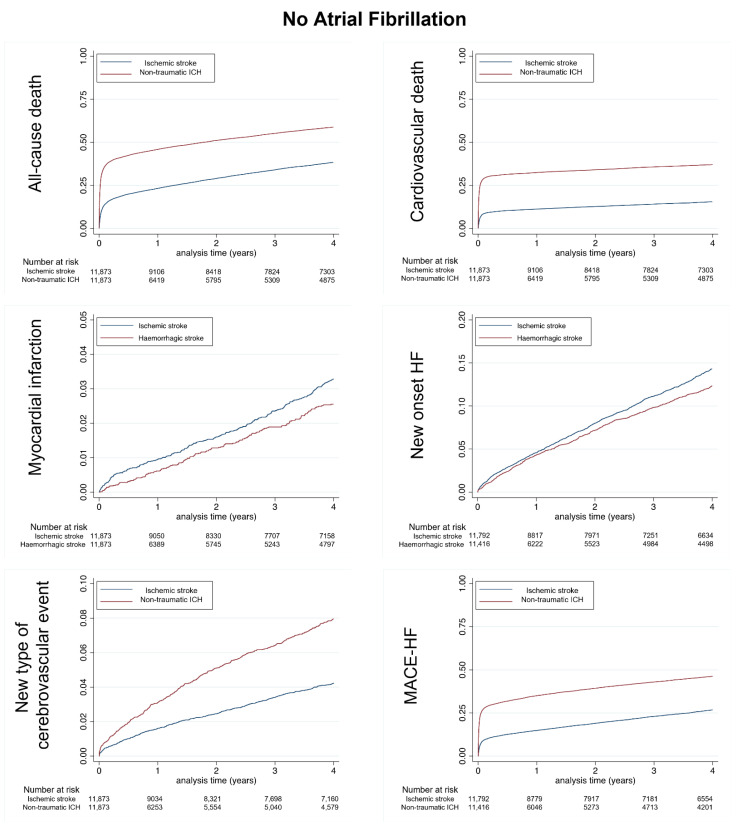
Cumulative incidences for all-cause death (**top left panel**), cardiovascular death (**top right panel**), myocardial infarction (**middle left panel**), new onset heart failure (**middle right panel**), new type of cerebrovascular event (ICrH for patients with history of ischemic stroke and ischemic stroke for patients with history of ICrH, **left lower panel)** and MACE-HF (**right lower panel**) during follow-up in the matched populations without comorbid AF. MACE-HF = Major cardiovascular events (in-hospital cardiovascular death, myocardial infarction, ischemic stroke or new-onset heart failure). ICH: non-traumatic intracranial hemorrhage.

**Figure 2 jcm-11-06885-f002:**
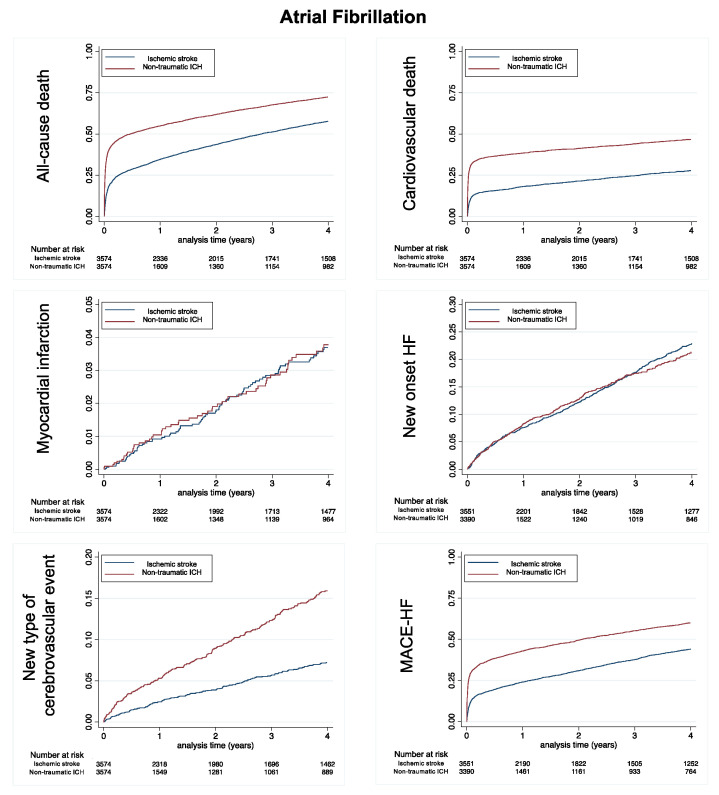
Cumulative incidences for all-cause death (**top left panel**), cardiovascular death (**top right panel**), myocardial infarction (**middle left panel**), new onset heart failure (**middle right panel**), new type of cerebrovascular event (non-traumatic ICrH for patients with history of ischemic stroke and ischemic stroke for patients with history of non-traumatic ICrH, **left lower panel**) and MACE-HF (**right lower panel**) during follow-up in the matched populations with AF. MACE-HF = Major cardiovascular events (in-hospital cardiovascular death, myocardial infarction, ischemic stroke or new-onset heart failure). ICH: non-traumatic intracranial hemorrhage.

**Table 1 jcm-11-06885-t001:** Baseline characteristics of unmatched patients with history and no history of atrial fibrillation at baseline in ischemic stroke and non-traumatic intracranial hemorrhage survivors.

	No Atrial Fibrillation		Atrial Fibrillation	
	Ischemic Stroke *N* = 4023	Non-Traumatic ICrH *N* = 12,028	Ischemic Stroke *N* = 20,449	Non-Traumatic ICrH *N* = 3574
Age (years), mean ± SD	72.3 ± 14.1	68.8 ± 17.1 ^#^	79.9 ± 9.7	79.4 ± 9.4 ^+^
Sex (male), *n* (%)	23,193 (57.2)	6430 (53.5) ^#^	10,350 (50.6)	1941 (54.3) ^#^
Hypertension, *n* (%)	26,772 (66.1)	6669 (55.4) ^#^	15,593 (76.3)	2739 (76.6) NS
Diabetes mellitus, *n* (%)	10,966 (27.1)	1995 (16.6) ^#^	5497 (26.9)	902 (25.2) NS
Smoker, *n* (%)	5535 (13.7)	1122 (9.3) ^#^	1490 (7.3)	227 (6.4) ^+^
Dyslipidemia, *n* (%)	13,885 (34.3)	2017 (16.8) ^#^	6820 (33.4)	916 (25.6) NS
Obesity, *n* (%)	5117 (12.6)	964 (8.0) ^#^	2894 (14.2)	490 (13.7) NS
Heart failure, *n* (%)	7014 (17.3)	1326 (11.0) ^#^	8810 (43.1)	1413 (39.5) ^#^
History of pulmonary edema, *n* (%)	568 (1.4)	182 (1.5)	739 (3.6)	100 (2.8) NS
Valve disease, *n* (%)	2494 (6.2)	364 (3.0) ^#^	3396 (16.6)	501 (14.0) ^#^
Aortic stenosis, *n* (%)	1074 (2.7)	169 (1.4) ^#^	1352 (6.6)	183 (5.1) ^§^
Aortic regurgitation, *n* (%)	559 (1.4)	62 (0.5) ^#^	676 (3.3)	111 (3.1) ^#^
Mitral regurgitation, *n* (%)	931 (2.3)	108 (0.9) ^#^	1612 (7.9)	233 (6.5) ^#^
Previous endocarditis, *n* (%)	221 (0.5)	61 (0.5) NS	209 (1.0)	57 (1.6) NS
Dilated cardiomyopathy, *n* (%)	1571 (3.9)	237 (2.0) ^#^	2024 (9.9)	293 (8.2) ^#^
Coronary artery disease, *n* (%)	7945 (19.6)	1281 (10.7) ^#^	6028 (29.5)	924 (25.9) ^#^
Previous MI, *n* (%)	1397 (3.4)	227 (1.9) ^#^	996 (4.9)	134 (3.7) ^#^
Previous PCI, *n* (%)	1514 (3.7)	243 (2.0) ^#^	873 (4.3)	131 (3.7) ^+^
Previous CABG, *n* (%)	242 (0.6)	26 (0.2) ^#^	295 (1.4)	34 (1.0) ^+^
Vascular disease, *n* (%)	10,907 (26.9)	1286 (10.7) ^#^	5915 (28.9)	720 (20.1) ^#^
Atrial fibrillation, *n* (%)	0 (0.0)	0 (0.0)	20,449 (100.0)	3574 (100.0)
Previous pacemaker or ICD, *n* (%)	1669 (4.1)	257 (2.1) ^#^	2790 (13.6)	432 (12.1) ^#^
Alcohol related diagnoses, *n* (%)	3643 (9.0)	1583 (13.2) ^#^	1232 (6.0)	285 (8.0) ^+^
Chronic kidney disease, *n* (%)	2367 (5.8)	527 (4.4) ^#^	2059 (10.1)	355 (9.9) *
Lung disease, *n* (%)	6853 (16.9)	2152 (17.9) ^+^	4766 (23.3)	880 (24.6) NS
Sleep apnea syndrome, *n* (%)	2167 (5.3)	409 (3.4) ^#^	1295 (6.3)	233 (6.5) NS
COPD, *n* (%)	3131 (7.7)	662 (5.5) ^#^	2095 (10.2)	370 (10.4) NS
Liver disease, *n* (%)	1440 (3.6)	748 (6.2) ^#^	754 (3.7)	184 (5.1) NS
Gastroesophageal reflux, *n* (%)	797 (2.0)	187 (1.6) ^§^	363 (1.8)	51 (1.4) *
Thyroid diseases, *n* (%)	2817 (7.0)	648 (5.4) ^#^	2711 (13.3)	524 (14.7) *
Inflammatory disease, *n* (%)	2370 (5.8)	521 (4.3) ^#^	1611 (7.9)	244 (6.8) *
Anemia, *n* (%)	4737 (11.7)	1215 (10.1) ^#^	3652 (17.9)	592 (16.6) ^+^
Previous cancer, *n* (%)	5520 (13.6)	1676 (13.9) NS	3035 (14.8)	540 (15.1) ^§^
Poor nutrition, *n* (%)	3232 (8.0)	978 (8.1) NS	2540 (12.4)	461 (12.9) *
Cognitive impairment, *n* (%)	5083 (12.5)	1430 (11.9) NS	3526 (17.2)	581 (16.3) NS
Illicit drug use, *n* (%)	208 (0.5)	70 (0.6) NS	31 (0.2)	5 (0.1) NS

Values are *n* (%) or mean ± SD. ICH = intracranial hemorrhage; SD = standard deviation; MI: myocardial infarction; PCI: percutaneous coronary intervention; CABG: coronary artery bypass graft; ICD: implantable cardioverter-defibrillator; COPD: chronic obstructive pulmonary disease; Comparisons are performed separately in patients with no atrial fibrillation and with atrial fibrillation; Non-significant = NS; *p*-value ≤ 0.05 *; *p*-value ≤ 0.01 ^+^; *p*-value < 0.001 ^§^; *p*-value < 0.0001 ^#.^

**Table 2 jcm-11-06885-t002:** Incident outcomes in the matched population with and without comorbid atrial fibrillation in ischemic stroke and non-traumatic intracranial hemorrhage survivors.

**No Atrial Fibrillation**	**Ischemic Stroke (*n* = 11,873)**		**ICrH (*n* = 11,873)**		***p* Value**
	**Number of Events**	**Incidence, %/Yr (95% CI)**	**Number of Events**	**Incidence, %/Yr (95% CI)**	
All-cause death	5758	12.71 (12.39–13.04)	7868	25.20 (24.65–25.76)	<0.0001
Cardiovascular death	1934	4.27 (4.08–4.46)	4351	13.94 (13.53–14.36)	<0.0001
Non-cardiovascular death	3824	8.44 (8.18–8.71)	3517	11.26 (10.90–11.64)	<0.0001
Myocardial infarction	474	1.06 (0.97–1.16)	266	0.86 (0.76–0.97)	0.006
New onset HF	1995	4.70 (4.50–4.92)	1171	3.97 (3.75–4.20)	<0.0001
MACE-HF	3713	8.84 (8.56–9.13)	5450	19.39 (18.88–19.91)	<0.0001
Major bleeding	2463	5.96 (5.73–6.20)	1595	5.58 (5.31–5.86)	0.04
New type of cerebrovascular event	591	1.32 (1.22–1.44)	705	2.36 (2.19–2.54)	<0.0001
**Atrial Fibrillation**	**Ischemic Stroke (*n* = 3574)**		**ICrH (*n* = 3574)**		***p* Value**
	**Number of Events**	**Incidence, %/yr (95% CI)**	**Number of Events**	**Incidence, %/yr (95% CI)**	
All-cause death	2511	23.89 (22.97–24.84)	2878	40.52 (39.06–42.03)	<0.0001
Cardiovascular death	966	9.19 (8.63–9.79)	1561	21.98 (20.91–23.09)	<0.0001
Non-cardiovascular death	1545	14.70 (13.98–15.45)	1317	18.54 (17.57–19.57)	<0.0001
Myocardial infarction	118	1.14 (0.95–1.36)	84	1.20 (0.97–1.48)	0.73
New onset HF	704	7.47 (6.94–8.04)	459	7.12 (6.50–7.80)	0.43
MACE-HF	1592	17.10 (16.28–17.97)	1946	32.31 (30.90–33.77)	<0.0001
Major bleeding	899	9.81 (9.19–10.47)	546	8.76 (8.05–9.52)	<0.0001
New type of cerebrovascular event	212	2.06 (1.80–2.35)	318	4.78 (4.28–5.33)	<0.0001

CI = confidence interval; HF = heart failure; ICrH = intracranial hemorrhage; MACE-HF = Major cardiovascular events (in-hospital cardiovascular death, myocardial infarction, ischemic stroke or new-onset heart failure).

**Table 3 jcm-11-06885-t003:** Hazard ratios (95% CI) associated with incident outcomes in non-traumatic intracranial hemorrhage with no atrial fibrillation (versus ischemic stroke survivors).

	Model A	Model B	Model C	Model D
All-cause death	1.76 (1.72–1.81)	2.05 (1.99–2.10)	1.89 (1.85–1.95)	1.80 (1.74–1.86)
Cardiovascular death	2.79 (2.68–2.89)	3.19 (3.08–3.32)	2.98 (2.86–3.09)	2.79 (2.64–2.94)
Non-cardiovascular death	1.23 (1.18–1.27)	1.43 (1.38–1.48)	1.31 (1.26–1.36)	1.28 (1.22–1.34)
Myocardial infarction	0.64 (0.56–0.73)	0.71 (0.62–0.81)	0.76 (0.67–0.87)	0.81 (0.70–0.94)
New onset HF	0.71 (0.67–0.76)	0.83 (0.78–0.888)	0.84 (0.79–0.89)	0.85 (0.79–0.91)
MACE-HF	1.83 (1.78–1.89)	2.11 (2.05–2.18)	2.06 (2.00–2.13)	1.97 (1.89–2.06)
Major bleeding	0.88 (0.83–0.93)	0.96 (0.91–1.01)	0.93 (0.88–0.98)	0.92 (0.87–0.98)
New type of cerebrovascular event	1.63 (1.49–1.78)	1.80 (1.65–1.95)	1.76 (1.61–1.92)	1.75 (1.57–1.96)

Model A: unadjusted; Model B: adjusted for age and sex; Model C: adjusted on all risk factors and non-cardiovascular comorbidities from Table 1; Model D: propensity score matched analysis. CI = confidence interval; HF = heart failure; MACE-HF = Major cardiovascular events (in-hospital cardiovascular death, myocardial infarction, ischemic stroke or new-onset heart failure).

**Table 4 jcm-11-06885-t004:** Hazard ratio ratios (95% CI) associated with incident outcomes in non-traumatic intracranial hemorrhage with AF (versus ischemic stroke survivors).

	Model A	Model B	Model C	Model D
All-cause death	1.53(1.47–1.59)	1.61 (1.55–1.68)	1.57 (1.51–1.63)	1.56 (1.47–1.64)
Cardiovascular death	2.06 (1.95–2.18)	2.19 (2.07–2.31)	2.16 (2.04–2.29)	2.08 (1.91–2.25)
Non-cardiovascular death	1.18 (1.11–1.25)	1.23 (1.16–1.30)	1.18 (1.11–1.25)	1.21 (1.13–1.31)
Myocardial infarction	0.99 (0.79–1.24)	0.99 (0.79–1.24)	0.99 (0.79–1.24)	1.05 (0.79–1.39)
New onset HF	0.92 (0.83–1.01)	0.95 (0.86–1.04)	0.96 (0.87–1.06)	0.94 (0.84–1.06)
MACE-HF	1.72 (1.64–1.80)	1.79 (1.70–1.88)	1.78 (1.69–1.87)	1.71 (1.60–1.83)
Major bleeding	0.88 (0.80–0.96)	0.88 (0.81–0.96)	0.86 (0.79–0.94)	0.89 (0.80–0.99)
New type of cerebrovascular event	2.28 (2.02–2.58)	2.32 (2.05–2.62)	2.29 (2.02–2.59)	2.29 (1.93–2.73)

Model A: unadjusted; Model B: adjusted for age and sex; Model C: adjusted on all risk factors and non-cardiovascular comorbidities from Table 1; Model D: propensity score matched analysis. CI: confidence interval; HF = heart failure; MACE-HF = Major cardiovascular events (in-hospital cardiovascular death, myocardial infarction, ischemic stroke or new-onset heart failure).

## Data Availability

Data are available upon reasonable request.

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
