# Peer review of "Adverse Cardiovascular Events in Non-Traumatic Intracranial Hemorrhage and Ischemic Stroke Survivors"

_jcm, 2022, doi:10.3390/jcm11236885_

Round 1
Reviewer 1 Report
The authors use the national registered data to extract the stroke patients with and without AF, then authors concluded that the ICrH survivors had higher cardiovascular events in the future. However, there are some points might need the help of authors to clarified:
1. Authors enrolled patients under hospitalization in 2013 with the ICD coded with IS, ICrH and AF, however, is this population diagnosed as first time stroke? or probably some of the cases could be second or even third stroke?
2. In the section of outcomes, authors evaluated the incidence of all-cause in-hospital death and cardiovascular death. however, what ICD-10 codes was used for cardiovascular death? or the ICD-10 codes used for cardiovascluar death is as same as major cardiovasculr events? is it possible to have a list of ICD-10 codes used for outcome in supplement data?
Author Response
The authors use the national registered data to extract the stroke patients with and without AF, then authors concluded that the ICrH survivors had higher cardiovascular events in the future. However, there are some points might need the help of authors to clarified:
- Authors enrolled patients under hospitalization in 2013 with the ICD coded with IS, ICrH and AF, however, is this population diagnosed as first time stroke? or probably some of the cases could be second or even third stroke?
Patients included indeed had a first-time stroke in the 2 years (2011-2012) preceding the inclusion in 2013. We cannot exclude that some patients had a previous stroke several years ago in a period not covered by the database. We do not think this may markedly affect our main results but this point is now included in the limitations.
- In the section of outcomes, authors evaluated the incidence of all-cause in-hospital death and cardiovascular death. however, what ICD-10 codes was used for cardiovascular death? or the ICD-10 codes used for cardiovascluar death is as same as major cardiovasculr events? is it possible to have a list of ICD-10 codes used for outcome in supplement data?
Mode of death (cardiovascular or non-cardiovascular) was identified based on the main diagnosis during hospitalization resulting in death based on ICD-10 codes (for cardiovascular death: I00–I99 – Diseases of the heart and circulatory system) as previously described [2 Ref].
The list of ICD-10 codes used for outcomes has been added to the supplement data.
1 - Fauchier L, Samson A, Chaize G, et al. Cause of death in patients with atrial fibrillation admitted to French hospitals in 2012: a nationwide database study. Open heart 2015; 2 :e000290.
2 - Deharo P, Bisson A, Herbert J, et al. Impact of Sapien 3 Balloon-Expandable Versus Evolut R Self-Expandable Transcatheter Aortic Valve Implantation in Patients With Aortic Stenosis: Data From a Nationwide Analysis. Circulation 2020; 141 :260-8.
Reviewer 2 Report
Many thanks for the opportunity to review your manuscript on "Adverse cardiovascular events in non-traumatic intracranial hemorrhage and ischemic stroke survivors." It reads well and brings an important message.
I suggest a minor revision:
- Introduction line 39/40: for clarity consider adding "subarachnoid haemorrhage constituted ~10%"
Author Response
Many thanks for the opportunity to review your manuscript on "Adverse cardiovascular events in non-traumatic intracranial hemorrhage and ischemic stroke survivors." It reads well and brings an important message.
We thank the reviewer for this supportive comment.
I suggest a minor revision:
- Introduction line 39/40: for clarity consider adding "subarachnoid haemorrhage constituted ~10%"
We have now added this clarification in the text.